# Malignant Knee Joint Effusion—A New Dimension of Laboratory Diagnostics

**Eliska Vanaskova** [1] [ID], **Petr Kelbich** [2,3,4] [ID], **Martin Cegan** [5] **and Tomas Novotny** [1,6,*] [ID]

1. Department of Orthopaedics, Masaryk Hospital, University J.E. Purkinje, 401 13 Usti nad Labem, Czech Republic; eliska.vanaskova@kzcr.eu
2. Department of Biomedicine and Laboratory Diagnostics, Masaryk Hospital, University J.E. Purkinje, 401 13 Usti nad Labem, Czech Republic; petr.kelbich@kzcr.eu
3. Department of Clinical Immunology and Allergology, Faculty of Medicine, University Hospital in Hradec Kralove, Charles University in Prague, 500 03 Hradec Kralove, Czech Republic
4. Laboratory for Cerebrospinal Fluid, Neuroimmunology, Pathology and Special Diagnostics Topelex, 190 00 Prague, Czech Republic
5. Department of Pathology, Masaryk Hospital, 401 13 Usti nad Labem, Czech Republic; martin.cegan@kzcr.eu
6. Department of Histology and Embryology, Second Faculty of Medicine, Charles University, 150 06 Prague, Czech Republic
* Correspondence: tomas.novotny@kzcr.eu; Tel.: +420-477113050

**Abstract:** Joint effusions are most frequently caused by osteoarthritis, trauma, an infection process or an autoimmune disease. The development of joint effusion due to a tumor process is rare but should be taken into consideration in the diagnostics. Joint effusions are examined mostly by means of microbiology to rule out or confirm pyogenic synovitis. These standard processes may take up to several days. The article presented here describes a unique case of a 74-year-old female diagnosed with a generalized malignant process according to a cytological-energy analysis and an immunocytochemical examination of a malignant joint effusion caused by femoral condyle metastasis. Other widely-used imaging methods such as X-ray, full-body CT scan and also laboratory examinations confirmed the malignancy and the origin. A cytological-energy analysis and an immunocytochemical examination can expedite the diagnostic process, can outline the processes happening in the joint and can indicate further examinations and subsequent therapy. The use of these laboratory methods appears to be a helpful diagnostic option to obtain additional information about a joint effusion, including the information about an ongoing malignant process. In our case report, they helped to confirm the typing of the tumor within three days, without the need for a metastasis biopsy. In appropriate cases, synovial fluid can play a role in tumor diagnostics.

**Keywords:** knee effusion; cytomorphology; cytological-energy analysis; immunohistochemical examination; joint effusion; malignancy; tumor diagnostics; orthopedics

## 1. Introduction

The synovium is an intra-articular mesenchymal tissue that is essential for the normal function of a joint [1]. Synovitis (i.e., inflammation of the synovial membrane) is a frequent condition (usually a secondary condition) caused by another ailment. The synovial processes can be posttraumatic inflammatory processes, e.g., infectious, parainfectious or autoimmune (due to rheumatoid arthritis, juvenile arthritis, sclerodermia, etc.), and mainly due to osteoarthritis. In the knee joint, causes of synovitis such as trauma, overuse, infections, gout, systemic causes, or changes in osteoarthritis, may cause the effusion [2,3]. However, tumors and tumor-like affections (such as chondroblastoma, osteoid osteoma, osteochondroma, synovial sarcoma, or metastatic processes) should not be overlooked [4]. Synovial fluid laboratory testing is an important part of a diagnostic evaluation of patients with joint diseases. Commonly, a specimen of synovial fluid is collected by arthrocentesis

and is analyzed macroscopically, biochemically, microscopically, and microbiologically. A macroscopic examination inspects the volume, the appearance, and the viscosity. Biochemically, the levels of glucose, lactate, coefficient of energy balance (KEB), aspartate aminotransferase (AST), uric acid, total protein, rheumatoid factor, etc., are identified. A microscopic analysis determines the cell count, differentials, and crystal identification. A microbiological examination is performed to detect microbes in the synovial fluid [5]. Our case report presented here describes a unique option in the diagnostic process using cytological-energy analysis as an accessible, rapid, and affordable method that reduces the necessity for further interventions.

## 2. Case Presentation

Our patient is a 74-year-old Caucasian female with a medical history of arterial hypertension, hyperlipidemia, and diabetes mellitus type 2. She was presented to the Emergency Department of Masaryk Hospital, Usti nad Labem, at the beginning of August in the night with severe pain in her left knee. She stated that the problems had already started in May. Since then, she had been treated in various orthopedic out-patient units with several corticosteroid injection applications into the left knee, and she had started using analgesics on a regular basis without significant relief. In the Emergency Room, blood tests, physical examination, and plain X-ray were carried out. The blood count and biochemical parameters came back without any pathologic values, only C-reactive protein (CRP) was detected as 8.9 mg/L. Our patient had all vital signs stable, her left knee was with effusion, and movements in the knee were limited due to severe pain. The X-ray (AP and lateral view) revealed focal changes in the femur condyles and ossification in the soft tissues of the knee (Figure 1), and an orthopedic examination was recommended.

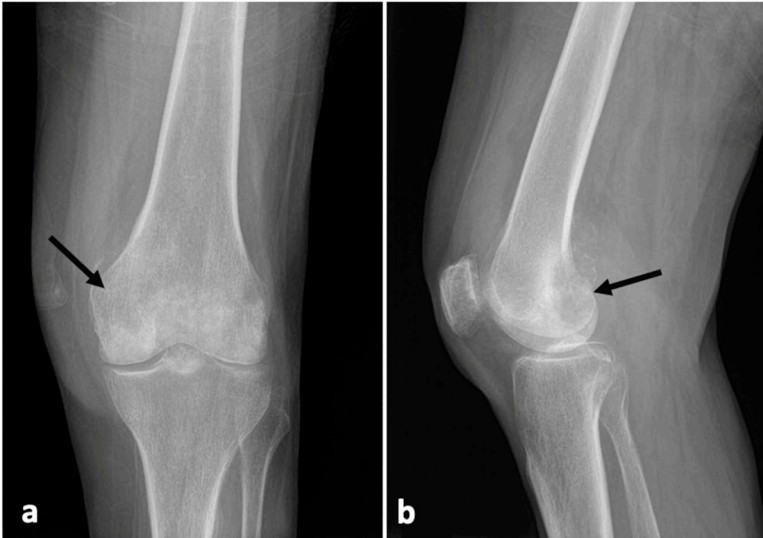

**Figure 1.** X-ray of the left knee: AP (**a**), lateral (**b**) views showing heterogenous remodeling of both condyles of the femur and irregular ossification in the soft tissue of the knee (black arrows).

The next day, the patient came to the examination room in our Orthopedic Department. During a physical examination, extension limitation to 20 degrees as well as limited flection to 70 degrees showed up. Rotations and abduction with adduction were normal. An ultrasound examination detected an effusion in the left knee, which was decided to be punctured for further examinations. The needle was inserted in the center of the synovial fluid in the suprapatellar recess guided by ultrasound and 15 milliliters of sanguinolent specimen was aspired. One half of punctured amount was sent for microbiological analysis, in which microbes were not detected. The second half was sent for a cytological-energy analysis to detect the presence of immunocompetent cells and their metabolic activity. We admitted the patient to our department for further examinations. During the admission

procedures, the patient revealed, in response to a direct question, that she had lost over 15 kg since the beginning of the knee problems in May. Furthermore, a suspicion of incipient dementia was detected. There was no known history of a breast cancer or any other types of cancer, neither in our patient's medical history nor in her family history. The patient underwent a CT scan of the left knee on the same afternoon. The conclusion from this examination was an osteolytic malignant process of the distal femur (Figure 2).

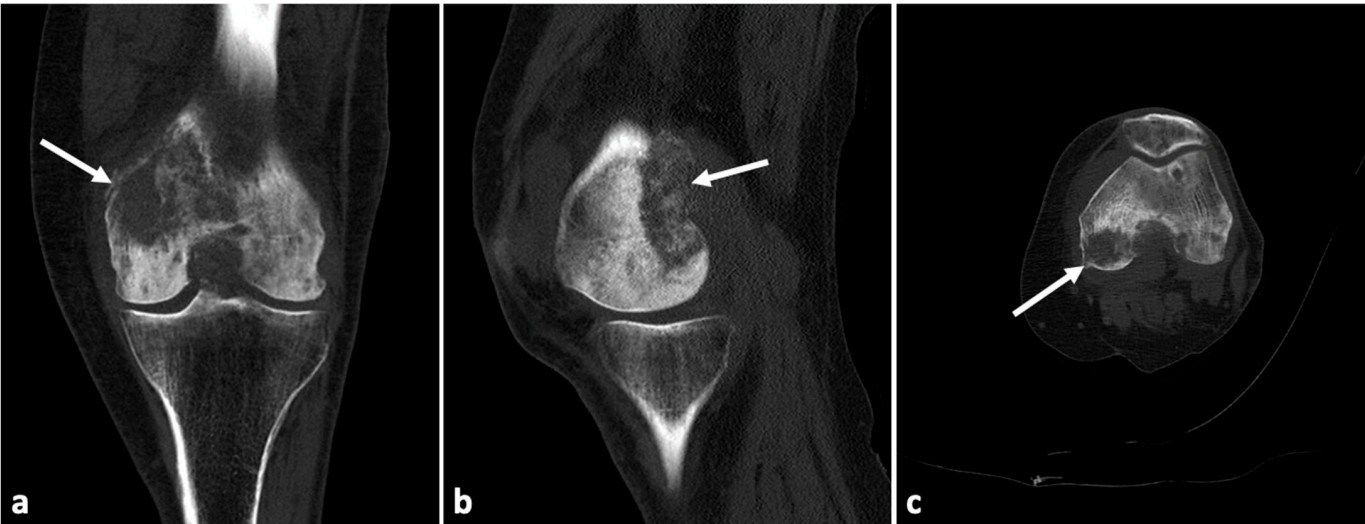

**Figure 2.** CT scan of the left knee: frontal view (**a**), sagittal view (**b**), transversal view (**c**), showing an osteolytic tumorous deposit in the left distal femur (white arrows).

Repeated blood tests came back without any abnormalities, just like the ones from the Emergency Room where only the CRP value was slightly higher with 9.0 mg/L. Meanwhile the results of a cytological-energy analysis returned with the interpretation of tumorous pleocytosis with the incidence of frequent malignant elements (Figure 3). In the effusion, immunocompetent cells were also present, predominantly neutrophils, with smaller amounts of lymphocytes and monocytes, and sporadic eosinophile granulocytes. Metabolic activity of the immunocompetent cells is presented by KEB. In addition, this analysis revealed a borderline elevation of CRP as an indicator of a slightly increased systematic inflammatory response. A mild anaerobic metabolism (KEB = 25.83) represented a local serous inflammatory response in the synovial space. An additional finding of a greatly elevated level of AST (=117.6 IU/L) indicated a tissue injury in the examined area. The cytology proved the presence of multiple malignant cells, which appeared as cells of adenocarcinoma (Figure 3a). The immunocytochemical analysis also revealed a positive expression of CK7 (Figure 3b) and mammaglobin (Figure 3c). The expression of other examined markers (TTF1, napsin A, CD56, ER, WT1, CA125, GATA3) was negative. The pathologist concluded that this was a case of generalized breast cancer.

Based on the outcomes of the cytological examination of the effusion, we started the process of oncological staging. A full-body CT scan was performed and it revealed multiple deposits of metastasis, mostly intracranial and some deposits in both of the adrenal glands, the left lung and liver, with malignant lymphadenopathy in several places across the body. The oncologist who was contacted required a blood test tumor markers' analysis. In the serum, the values of the tumor markers were AFP (9.64 kIU/L), CA 19-9 (195.9 kU/L), CA 15-3 (15.7 kU/L), CEA (4.12 μg/L), NSE (30.24 μg/L), Beta HCG (1.55 IU/L). Based on these findings and the extent of the disease, the oncologist recommended only symptomatic therapy without any further examinations. The palliative team was contacted and started to provide transport to a long-term palliative facility.

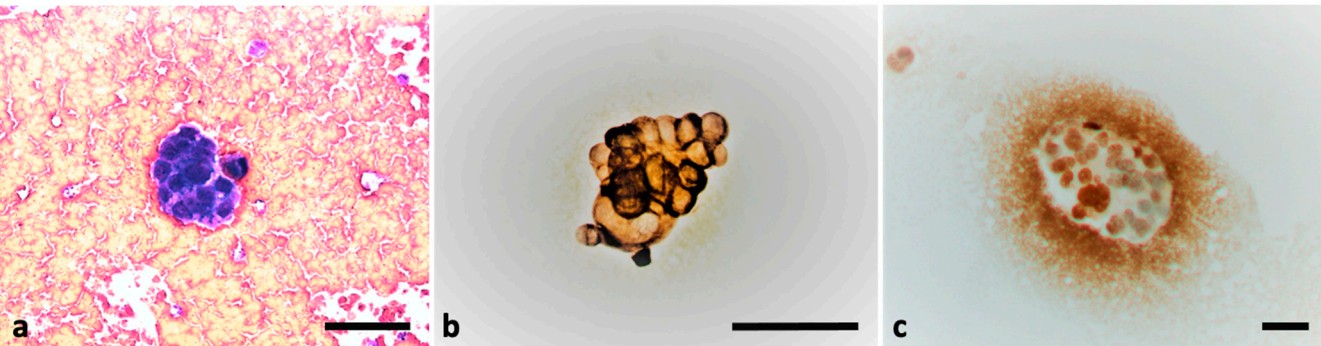

**Figure 3.** The presence of malignant elements in the effusion viewed in the cytological analysis: Metastatic breast adenocarcinoma cells in synovial fluid stained with Hemacolor (Merck Co., Germany) (**a**), CK7-expressing metastatic breast adenocarcinoma cells in the synovial fluid stained using immunocytochemistry (Dako, USA) (**b**), mammaglobin-expressing metastatic breast adenocarcinoma cells in synovial fluid stained using immunocytochemistry (Zytomed Systems GmbH, Germany) (**c**). Scale bar = 50 μm.

## 3. Discussion

Joint pain and joint effusion are two symptoms that every orthopedic surgeon comes across on a daily basis. These symptoms are mainly caused by overuse, osteoarthritis, trauma or infections, and by autoimmune diseases [3]. The development of malignant joint effusion in patients with solid tumors is infrequent. However, this potential cause should be taken into consideration, as it may help in the diagnosis of the origin of the tumor.

Around one quarter of deaths in the developed world are from cancer [6]. Bone metastases are responsible for high morbidity in cancer patients and their incidence is still increasing. These metastases are also the third most common metastatic site in patients with advanced cancer [7]. Metastases from carcinomas are among the most frequent malignant tumors involving bone. They are most frequently received from the breast, the prostate, and the gastrointestinal tract [8]. About 25% of these patients remain asymptomatic, and in the remaining 75%, bone metastases are responsible for various clinical complications, such as pain, pathological fractures, etc. [9].

Metastatic joint and synovium disease is one of the rarest manifestations of malignant disease, with only 37 cases previously reported in the literature [10,11]. The knee is the most commonly affected joint, ahead of the shoulder, hip, sternoclavicular, interphalangeal joint, wrist, and ankle [12]. Polyarticular involvement is rare. Solid tumors were diagnosed in 34 cases, in which adenocarcinoma was the main histopathological type [11]. In some reported cases, swelling and effusion of the joint appeared in patients with a known tumor diagnosis in their personal history [13].

Knee joint aspiration is performed to aid in the diagnosis and treatment of knee joint diseases [14]. A cytologic evaluation of the synovial fluid was performed in only one third of the cases reported in the literature. Tumor cells were presented in only one half of the cases. The sensitivity, therefore, appeared to be low, as the absence of tumor cells does not rule out a tumor invasion [11,12].

A synovial biopsy or an arthroscopy resection of the synovium was performed in earlier reported cases. In our case, there was no reason to perform additional interventions as we had implemented the cytological-energy analysis followed up by an immunocytochemical analysis by the pathologist. The method that we are discussing in our case report —cytological-energy analysis of knee effusion allows us to determine the type and intensity of a local or an inflammatory response in the joints. The focal inflammatory response is not specific to tumor involvement. Nonetheless, it provides important information about the state of local immunity, the development of the pathological process and infectious or other complications. In the case of our patient, KEB value of 25.83 revealed a local pathological process with a mild serous inflammatory response and excluded purulent

complications caused by bacteria in the knee joint. Our recent studies concluded KEB values from 38.0 to 28.0 indicate a predominance of aerobic metabolism in cases of absence of very mild inflammatory reaction. KEB values from 28.0 to 20.0 demonstrate a slight increase in anaerobic metabolism in cases of mild serous inflammatory reaction. KEB values from 20.0 to 10.0 are an indication of a slight increase in anaerobic metabolism in cases of a more severe serous inflammatory reaction or intense inflammation with an oxidative burst of professional phagocytes. Lastly, KEB values bellow 10.0 demonstrate strong anaerobic metabolism in cases of very intense inflammation with oxidative burst of professional phagocytes, e.g., purulent inflammation induced by extracellular bacteria [15–20].

The cytological part of the cytological-energy analysis detected the presence of malignant elements of the effusion. The subsequent immunocytochemical analyses enabled an early typing of the tumor and thereby, an early diagnosis without necessity of further interventions. Accompanied by other symptoms, a physical examination and further imaging methods in combination may expedite the process of specifying the origin of the tumor and may accelerate the beginning of targeted therapy. A cytological-energy analysis of extravascular body fluids, including synovial fluid, also determines the type of local inflammatory response in the relevant area [17]. The usual immune response to tumor is a cytotoxic reaction, antibody-mediated inflammation, or a Th1 lymphocyte-driven reaction with an oxidative burst of macrophages [18]. A predominance of lymphocytes is therefore typical for malignant effusions [21]. In the case of our patient, we observed a predominance of neutrophils and mild anaerobic metabolism in her synovial fluid. We evaluated this finding as a significantly increased risk of purulent inflammatory complication in the knee [19]. Tumors with synovial metastasis seem to be very aggressive with a disseminated disease at the time of diagnosis. The prognosis is poor, with the average survival being less than 5 months [22]. The suggested treatment in the reported cases is palliative care such as systematic chemotherapy or local analgetic radiation, and in more generalized patients, as in our case, only analgotherapy.

The possibility of metastatic disease should be taken into consideration when a patient with a history of previous malignancy presents with a chronic arthritis [11].

## 4. Conclusions

A cytological-energy analysis is a rapid and affordable method for which the specimen may be obtained in any orthopedic practice by puncturing the effusion. This method may be used to expedite the process of diagnosis and the beginning of treatment, and to avoid exposing the patient to more invasive methods leading to the origin of the cancer.

**Author Contributions:** Writing—original draft preparation and conceptualization, E.V., P.K., M.C. and T.N.; Supervision, T.N.; funding acquisition, T.N. All authors have read and agreed to the published version of the manuscript.

**Funding:** This work was supported by Krajska zdravotni a.s., Usti nad Labem, Czech Republic (No. IGA-KZ-217116003) and by Faculty of Health Studies, University J.E. Purkinje, Usti nad Labem, Czech Republic.

**Institutional Review Board Statement:** This was a purely observational case study that did not alter the patient's management and clinical outcomes. Thus, ethical approval was not required for this case report.

**Informed Consent Statement:** Informed consent was obtained from the patient involved in the study. Written informed consent has been obtained from the patient to publish this paper.

**Data Availability Statement:** The data that support the findings of this study are available from the corresponding author upon reasonable request.

**Conflicts of Interest:** The authors declare no conflict of interest.

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
