# Peer review of "Malignant Knee Joint Effusion—A New Dimension of Laboratory Diagnostics"

_applsci, doi:10.3390/app12030994_

Round 1
Reviewer 1 Report
Comments to the Author:
The authors reported a case in which a cytological-energy analysis was useful for a cancer diagnosis in a patient with synovial metastasis. There are some concerns of this report as written that a cytological-energy analysis plays a key role of an early diagnosis with cancer.
Specific queries are below:
- There is too little information about cytological-energy analysis, the core of this case report. As the author mentioned in this case, KEB=25.83 value at synovial joint fluid could be considered as a result of inflammatory response. The diagnosis of cancer was confirmed by cytology from aspirated synovial fluid. The causal relationship was mentioned unclearly between KEB value and cancer etiology. Isn’t this a value which could come from other inflammatory responses as well? If cytological-energy analysis detects inflammation in the synovial fluid, what are the benefits of this additional test compared the conventional joint fluid analysis? It seems that further explanation is needed. Similarly, what is the source for the definition of KEB value? A reference is needed.
page 3 - “A mild anaerobic metabolism (KEB = 25.83) represented a local serous inflam- matory response in the synovial space. This detection was based on the etiology of the tumor.”
- Please describe how much volume was aspirated. Even if there is metastatic synovial effusion, there will be the cases of undetected cancer cells due to dilutions or other inflammatory materials. This could give information how much volume of synovial fluid can be used for cancer analysis. Where was the needle tip targeted during sono-guided aspiration?(the center of synovial fluid, or as close to the articular cartilage?)
- This method may not be applied to all tumor types. But, in case of stage Ⅳ with metastatic focus, this method might have a certain novelty value to diagnose the patient’s status. However, like the patient in this case, this method is unlikely to affect the further cancer treatments. (most of patients who could be applied this method are provided a palliative care) What do the authors think the usefulness of this method will be on patient prognosis?
- Please describe laboratory findings and vital signs at the ER.
Reviewer 2 Report
The authors described a case study where cytological-energy analysis was performed on knee effusion alongside X-ray and CT scans upon initial examination. The cytological-energy analysis and immunocytochemistry results were used to diagnose the patient with a metastatic breast cancer without the traditional metastatic biopsy and further helped to inform the subsequent therapy regime. While the overall case study was novel and presented clearly, a few comments need to be addressed.
- A brief review of cytological-energy analysis should be included in the introduction. This review should go over the basic principles and outcomes this this analysis, the current use case of this analysis in knee effusion, and the benefits of more broadly applying this analysis.
- Need to highlight the important features in CT scan, e.g. arrows pointing to the tumorous deposits.
- Need to show scale bars for Figure 3.
- It's unclear how the results of cytological-energy analysis inform oncological staging. Further, the results of staging was not clearly stated.
- There are multiple language errors that require further editing and correction. Line 81: Incomplete sentence. Line 146: spelling error, etc.
Reviewer 3 Report
In the manuscript entitled “Malignant knee joint effusion – a new dimension of laboratory diagnostics” by Eliska Vanaskova et al., presents a malignant joint effusions secondary to breast tumor metastasis. The authors underlined cytological-energy analysis and IHC examination accelerated the diagnostic process after X-ray and full-body CT scan confirmed the malignancy. Through malignant joint effusion is not an unusual disorder and cytological-energy analysis is not a challenging differential diagnosis or improved technical procedures. But the theme may be interested for readers.
1. The authors could be clear denoting what kind of appropriate cases fit the procedures.
2. The authors should include the breast cancer history of this patient.
3. Have any different in cytological-energy analysis result of case with synovial sarcoma or synovial metastasis?
Round 2
Reviewer 1 Report
Comments to the Author:
- Abstract (line 27-29)
The use of these laboratory methods confirmed the malignant process and confirmed the typing of the tumor within three days without the need for a metastasis biopsy.
- Even after checking the reference in Author’s comments, this sentence seems to go too far. Although the analysis could be helpful to obtain additional information, it cannot say the cytology analysis “confirms” tumor typing or malignant process. This sentence needs to be corrected.
- Author’s comments에서 reference를 확인 후에도 이 문장은 무리가 있어 보입니다. Additional information을 얻을 수 있다는 정도이지 아직은 cytology analysis가 tumor typing이나 malignant process를 confirmation한다고 말하기에는 무리가 있어 보입니다. 문장 수정이 필요해 보입니다.
- Introduction (line 53)
Exact typing of the tumor by analyzing the effusion
- It seems unreasonable to mention that “exact typing of the tumor by analyzing the effusion”. This sentence needs to be corrected.
- Case presentation (line 101,102)
KEB is defined as the theoretical average number of ATP molecules produced from one glucose molecules produced from one glucose molecule under given conditions in a relevant compartment.
- Adding a reference might be enough. The sentence in that section seems to be redundant, and it would be better to delete this sentence.
Author Response
Please see the attachment.

This manuscript is a resubmission of an earlier submission. The following is a list of the peer review reports and author responses from that submission.